# VLP-Based COVID-19 Vaccines: An Adaptable Technology against the Threat of New Variants

**DOI:** 10.3390/vaccines9121409

**Published:** 2021-11-30

**Authors:** Wasim A. Prates-Syed, Lorena C. S. Chaves, Karin P. Crema, Larissa Vuitika, Aline Lira, Nelson Côrtes, Victor Kersten, Francisco E. G. Guimarães, Mohammad Sadraeian, Fernando L. Barroso da Silva, Otávio Cabral-Marques, José A. M. Barbuto, Momtchilo Russo, Niels O. S. Câmara, Gustavo Cabral-Miranda

**Affiliations:** 1Department of Immunology, Institute of Biomedical Sciences, University of São Paulo (ICB/USP), São Paulo 05508000, SP, Brazil; wasim.syed@usp.br (W.A.P.-S.); Karincrema@usp.br (K.P.C.); vuitika@usp.br (L.V.); aline.llira@usp.br (A.L.); nelson.cortes@usp.br (N.C.); vkersten@biof.ufrj.br (V.K.); otavio.cmarques@usp.br (O.C.-M.); jbarbuto@icb.usp.br (J.A.M.B.); momrusso@usp.br (M.R.); niels@icb.usp.br (N.O.S.C.); 2Institute of Research and Education in Child Health (PENSI), São Paulo 01228200, SP, Brazil; 3Department of Microbiology and Immunology, School of Medicine, Emory University, Claudia Nance Rollins Building, Atlanta, GA 30329, USA; lorenachaves@emory.edu; 4São Carlos Institute of Physics, IFSC-USP, São Carlos 13566590, SP, Brazil; guimarae@ifsc.usp.br (F.E.G.G.); msadraeian@usp.br (M.S.); 5Institute for Biomedical Materials & Devices (IBMD), Faculty of Science, University of Technology, Sydney, NSW 2007, Australia; 6Department of Biomolecular Sciences, School of Pharmaceutical Sciences of Ribeirão Preto, University of São Paulo, Ribeirão Preto 14040903, SP, Brazil; flbarroso@usp.br; 7Department of Chemical and Biomolecular Engeneering, North Carolina State University, Raleigh, NC 27695, USA; 8Department of Clinical and Toxicological Analyses, School of Pharmaceutical Sciences, University of São Paulo, São Paulo 05508000, SP, Brazil; 9Network of Immunity in Infection, Malignancy and Autoimmunity (NIIMA), Universal Scientific Education and Research Network (USERN), Children’s Medical Center, Tehran 1419733151, Iran; 10Laboratory of Medical Investigation in Pathogenesis and Targeted Therapy in Onco-Immuno-Hematology (LIM-31), Department of Hematology, Hospital das Clínicas HCFMUSP, Faculdade de Medicina, Universidade de São Paulo, São Paulo 0124690, SP, Brazil

**Keywords:** virus-like particles, vaccines, COVID-19, SARS-CoV-2

## Abstract

Virus-like particles (VLPs) are a versatile, safe, and highly immunogenic vaccine platform. Recently, there are developmental vaccines targeting SARS-CoV-2, the causative agent of COVID-19. The COVID-19 pandemic affected humanity worldwide, bringing out incomputable human and financial losses. The race for better, more efficacious vaccines is happening almost simultaneously as the virus increasingly produces variants of concern (VOCs). The VOCs Alpha, Beta, Gamma, and Delta share common mutations mainly in the spike receptor-binding domain (RBD), demonstrating convergent evolution, associated with increased transmissibility and immune evasion. Thus, the identification and understanding of these mutations is crucial for the production of new, optimized vaccines. The use of a very flexible vaccine platform in COVID-19 vaccine development is an important feature that cannot be ignored. Incorporating the spike protein and its variations into VLP vaccines is a desirable strategy as the morphology and size of VLPs allows for better presentation of several different antigens. Furthermore, VLPs elicit robust humoral and cellular immune responses, which are safe, and have been studied not only against SARS-CoV-2 but against other coronaviruses as well. Here, we describe the recent advances and improvements in vaccine development using VLP technology.

## 1. Introduction

The SARS-CoV-2 (Severe Acute Respiratory Syndrome Coronavirus 2) is the causative agent of COVID-19 (Coronavirus Disease 2019) [1,2,3] and is responsible for the recent pandemic, which has already reported 248,467,363 cases and 5,027,183 deaths worldwide as of 5 November 2021 [4]. As new cases continue to increase worldwide, it is urgent to develop inexpensive and versatile vaccines to handle emerging variants that can affect pre-existing natural immunity and the efficacy of already approved vaccines [5]. According to the World Health Organization, 129 COVID-19 vaccines are under clinical trials, and eight are approved for emergency or definitive use worldwide, including inactivated, mRNA and viral vector vaccines (Table 1) [6,7]. Although we already have these available vaccines, there is still a need for improved versions of COVID-19 vaccines. Hence, adapting vaccines to variants of concern (VOCs) along with decreasing vaccine costs will be the goals for the next step towards eradication. A technology that has potential to address some of these issues is the virus-like particles (VLPs) vaccine platform, as it is adaptable, resembles viral structures, highly immunogenic, and can be less expensive than other platforms.

The VLPs are noninfectious nanoscale particles composed of single or multiple self-assembling viral or nonviral proteins, which mimic a native viral particle [27]. These particles, when used as immunogens, are captured and processed by antigen presenting cells (APCs) and presented by both MHC-I and MHC-II to T helper and Cytotoxic T lymphocytes (Figure 1A). The structural repetitiveness and particle size of VLPs enhance recognition and direct activation of B cells. [28,29,30,31]. Taken altogether, these properties lead to robust humoral and cellular immune responses, which are exciting for vaccination against infectious diseases [32,33,34,35,36,37,38].

VLPs are classified according to their structural composition (nonenveloped-neVLPs or enveloped-eVLPs) and to the native virus components (homologous or heterologous) [39] (Figure 1B). Homologous VLPs comprise particles that self-assemble using proteins derived from the native virus only [40]. On the other hand, heterologous VLPs contain proteins from different sources to increase immunogenicity [41]. Moreover, available bioinformatics tools can help to optimize the rational design of new and pre-existing VLPs to achieve the best immunogenic performance [32,42,43,44,45,46].

VLPs can be expressed in insect [32,47], mammalian [48,49], bacterial [50], plant [37,51] or yeast cells [33]. These particles have been used for the development of vaccines against several infectious diseases [52]. VLP-based vaccines for Hepatitis B (Engerix-B^®^), Hepatitis E (Hecolin^®^) and HPV (Cervarix^®^, Gardasil^®^ and Gardasil 9^®^) demonstrate proof of concept for industrially scalable vaccines [33,53,54]. In addition to vaccine development, the empty shell-like structure of these nanoparticles has been studied for other biomedical purposes, such as gene therapy [53,54,55,56] and drug delivery [57,58,59,60,61,62].

## 2. SARS-CoV-2, VOCs, and Structural Vaccinology

The SARS-CoV-2 positive-sense single-stranded RNA genome (29 kb in length) encodes four structural and 16 non-structural proteins [3]. The structural proteins are the membrane (M), envelope (E), spike (S), and nucleocapsid (N) proteins, as seen in other coronaviruses (Figure 2A). The M protein (UNIPROT ID: P0DTC5) is a highly conserved homodimer and is the most abundant protein in mature virions [43,44]. It plays an essential role in viral assembly, membrane fusion, and particle size and shape, through interactions with other structural proteins [63,64,65,66]. The E protein (UNIPROT ID: P0DTC4) is a pentameric structure [67] found at low levels among coronaviruses [68,69]. It regulates M homodimerization [70] and, along with M, plays a central role in the vesicular transport of virions [71,72,73], ion transport [74,75,76,77,78], and pathogenesis [79,80,81]. The N protein (UNIPROT ID: P0DTC9) is a highly positive-charged, flexible, and unstable [58] phosphoprotein [82,83] that essentially performs genomic packaging [48,84,85]. The N-terminal domain of this protein [86,87] interacts with the viral RNA genome, and the C-terminal domain [84] forms a helical-shaped nucleocapsid [66,88,89]. Overall, the N protein interactions with M and viral RNA define the shape and size of the virions [85]. Unsurprisingly, N is essential in early viral replication and plays an integral role in infectivity [90].

Among these structural proteins, the main target for vaccine development is the SARS-CoV-2 S protein, which gives the characteristic crown-shaped structure of coronaviruses [88,89]. S is a highly glycosylated [93] homotrimer transmembrane protein (UNIPROT ID P0DTC2) composed of 1273 amino acids per chain. Known human sarbecoviruses (SARS-CoV-2 and SARS-CoV-1) and the alphacoronavirus NL63 invade the host cell through an interaction between the S protein and its receptor, the angiotensin-converting enzyme 2 (ACE2) [93,94,95,96,97] (Figure 2B). In general, the S protein consists of two major regions in addition to the signal peptide (SP) (1–12): the S1 subunit (13–685), and the S2 subunit (686–1273), which contains the transmembrane region (TM) (1214–1234) followed by the cytoplasmic domain (CD) (1235–1273) (Figure 3A). The S protein and the ACE2 receptor binding are mediated by the receptor-binding motif (RBM; 437–508), located in the receptor-binding domain (RBD; 319–541) [98] (Figure 3A, purple and cyan, respectively). The fusion machinery in S2 is composed of two fusion peptides (816–837 and 835–855) and two heptad regions (920–970 and 1163–1202). The first site of cleavage targeted by host proteases, such as furin and TMPRSS2, is located in the S1/S2 interface (685–686) [99,100,101] (Figure 3A, red)**.** Removing the S1/S2 site promotes conformational changes that open the second cleavage site at S2 (815–816). The subsequent cleavage of the S2 site promotes the projection of needle-shaped fusion peptides into the host membrane [102,103], leading to cell fusion in 60–120 s in feline coronavirus [104]. The S protein presents a closed and open conformation [45,105] (Figure 3B, upper and bottom panel, respectively). With one or more RBDs projected outward, the open state constitutes the major conformation population of viable virions [105]. The increased exposure and steric freedom enable stronger interactions with the ACE2 receptor [45,106]. Therefore, mutations that stabilize this open conformation lead to positive selection, making the virus more transmissible [107,108,109].

A better understanding of the effects of these conformational changes allows us to closely monitor the emerging VOCs. The VOCs that have attracted the most attention so far are the Alpha, Beta, Gamma, and Delta variants [108], which were initially identified in the UK [109], South Africa [112], Brazil [113], and India [114], respectively (Figure 4).

The Alpha variant (B.1.1.7; main mutations in RBD: N501Y, P681H, 69/70 Δ) was first identified in the United Kingdom and was also the first VOC to receive global attention due to increased transmissibility up to 43–90% higher than the original strain [112,115]. Furthermore, the Alpha variant was associated with increased disease severity among younger patients [116]. Nevertheless, this VOC did not significantly affect the efficacy of available vaccines [117,118]. The Beta variant (B.1.351; main mutations in RBD: E484K, K417N, N501Y) was first detected in the Eastern Cape province of South Africa and was associated with decreased vaccine efficacy [18,112] and higher immune evasion than the Alpha variant [119,120]. The Gamma variant (P.1; main mutations in the RBD: K417T, E484K, N501Y, D614G) has become the primary concern in Brazil, as it is 1.7–2.4 fold more transmissible compared to the Alpha variant [113]. Indeed, the Gamma variant spread rapidly in the Amazonas, Brazil, and contributed to the collapse of the local health care system [121]. Interestingly, both the Gamma and Alpha variants have similar binding affinities between the RBD and ACE2, slightly stronger than the original strain [105]. Stronger receptor binding could explain the higher viral transmission observed in countries with the dominant circulating Gamma variant [122,123]. As mentioned above, these variants have contributed to the collapse of the public health system in many countries. However, recently, a new emerging variant (Delta) has become the highest global health concern as it was shown to affect immune responses [18,124,125] and is 55% more contagious than the Alpha variant [126]. The Delta variant (main mutations: T19R, G142D, E156G, F157Δ, R158Δ, L452R, T478K, D614G, P681R, and D950N) was first identified in India in December 2020 [114,127] and quickly became the local predominant VOC. The accumulation of convergent mutations on VOCs resulted in local variants during the COVID-19 s wave in India. As expected, due to the increased transmissibility of Delta, this variant soon became the predominant VOC in England [109], the United States [128,129], and other countries [18]. The linages of VOCs are defined by multiple convergent mutations that are hypothesized to have appeared in the condition of chronic COVID-19 infections in immunocompromised patients [130]. A recent study showed that patients who received the first dose of the BNT162b2 (Pfizer-BioNTech) and ChAdOx1 AZD1222 (Oxford-AstraZeneca) vaccines were not as protected against the Delta variant as those who received the second dose [131].

The VOCs have become a threat to pandemic control, and the vaccines need the help of biotechnology, especially structural vaccinology tools, to stay one step ahead of emerging variants. Structural vaccinology proposes the rational design and selection of antigens, including peptides, to maximize vaccine immunogenicity, safety, stability, and quality [43,132]. This critical process can accelerate the development of new vaccines through several bioinformatics, biophysical, and computational tools already available [133,134,135,136,137]. Such resources offer a rational approach to predict epitopes and measure their binding affinities and structural and dynamical properties [138,139].

Different laboratories have already reported potential B and T cell epitopes predicted based on the SARS-CoV-2 S protein that could be used as vaccine candidates, including VLPs vaccines [138,140,141]. For example, the UB-612 is a rationally designed vaccine candidate construct based on two antigenic components: an S1-RBD-sFc fusion protein and a synthetic polypeptide consisting of five SARS-CoV-2 derived peptides together with a universal peptide [132]. This alum- and CpG-adjuvanted vaccine candidate induced high titers of S1-RBD nAbs and increased Th1-oriented responses. In mice, UB-612 reduced the viral load and prevented the development of the disease in a live SARS-CoV-2 challenge. UB-612 is being developed by Vaxxinity, which is currently conducting clinical trials in Taiwan. Another successful example employing structural vaccinology is the use of a well-known structural optimization that improves the expression and stability of the S protein in MERS-CoV and was adapted to SARS-CoV-2 protein-based vaccines [45]. Two consecutive substitutions of proline residues at 986–987 on the S2 subunit (S2P) and mutations at the cleavage site S1/S2 (682–685, ‘RRAR’ to ‘GSAS’) increased protein expression and prevented host proteolytic cleavage, locking the S protein in the prefusion state [45,142]. The S2P mutations are being used in the available Pfizer-BioNTech vaccine (Comirnaty^®^ BNT162b2) [25], which showed high efficacy (95%) [26] against symptomatic cases and a substantial reduction of hospitalizations and deaths in the United Kingdom, Israel, and other countries [143,144,145]. Other studies have shown that point mutations at residues T572I, D614N, A892P, A942P and V987P lock S in the closed state and promote a 6.4-fold increase in protein expression and thermal stability [115]. Despite the expression advantages, these mutations could make vaccines more accessible to developing countries [146].

## 3. Enveloped VLPs against SARS-CoV-2

As discussed above, incredible progress has been made on the understanding and improvement of SARS-CoV-2 immunogens. Significantly, the comprehensive analysis of the S protein characteristics and its antigenic regions were crucial for the rapid development of the SARS-CoV-2 vaccines. However, there is still room to test both existing and novel antigens using different platforms.

The VLP vaccine platform can be used for the production of eVLPs and neVLPs of interest [140,147]. Enveloped VLPs harbor a host-derived or synthetic membrane and generally need more complex expression systems, such as those in eukaryotic cell lines (mammalian, insect, or plant cells). For the formation of coronavirus eVLPs, structural proteins are essential for viral maturation and particle assembly [48,49,148]. Some studies have shown that the minimal requirement for the assembly of SARS-CoV-2 VLPs and other coronaviruses is the combination of M and either the E or N proteins. However, most particles include the N protein and the highly immunogenic S protein for better assembly and expression (Figure 5A) [48,49]. To date, Vero E6 cells presented the highest expression of S-containing VLPs when compared to HEK293 cells [49]. All of these initial approaches for SARS-CoV-2 VLP production show a promising use of this platform in vaccine development. However, industrial viability and large-scale production was not considered, and these are crucial features for further development and are still very challenging in the eVLP production field [100].

Although homologous VLPs are an attractive strategy for producing these particles, the combination of antigenic SARS-CoV-2 proteins with other highly expressed heterologous proteins (that could be used as alternative VLPs scaffolds) are an exciting strategy to address issues of industrial production. The NDVLP-S2P (Figure 5B) is a heterologous chimeric eVLP vaccine candidate against SARS-CoV-2 that uses the structure of a well-characterized enveloped virus, the Newcastle disease virus (NDV) [41], and is being developed by the National Institute of Allergy and Infectious Diseases (NIAID). The transmembrane domain of the NDV fusion protein was fused to SARS-CoV-2 S2P, allowing the correct display of S2P on the VLP surface [41,45,149]. The NDV-S2P VLPs were more immunogenic than the trimeric S protein alone, showing that the presentation of antigens on the surface of the VLPs is advantageous [41]. Another heterologous SARS-CoV-2 eVLPs vaccine candidate is the CoVLP from Medicago/GSK (Figure 5C) [37,51]. This vaccine is based on VLPs that display a mutated S2P protein, which comprises a plant signal peptide, GSAS substitutions in the S1/S2 site, and TM/CD regions of the Influenza H5 A/Indonesia/5/2005. The CoVLP vaccine is formulated with AS03 [141] and given in a two-dose regimen. After the second dose, immunized volunteers showed higher serum SARS-CoV-2 nAb titers than in convalescent plasma. This vaccine candidate is already in phase 3 clinical trials (NCT04636697). VBI-2902a [149] is an MLV-based eVLPs vaccine candidate containing the S protein in the prefusion state fused with the VSV-G transmembrane cytoplasmic domain (VSV-GTC) (Figure 5D). This vaccine is being developed by VBI Vaccines and is in ongoing clinical trials 1/2 (NCT04773665).

As discussed, all these approaches allow for easier incorporation of highly antigenic S proteins into the VLP scaffold, dramatically improving production and efficacy of the vaccine.

## 4. Non-Enveloped VLPs against SARS-CoV-2

Unlike eVLPs, neVLPs do not contain any lipid membranes. They can be produced in simpler expression systems, such as those using bacteria (i.e., *Escherichia coli*) and yeast (i.e., *Saccharomyces cerevisiae* and *Pichia Pastoris*) cells. The Hepatitis B virus vaccine, Engerix-B^®^ [150], and the Human papillomavirus vaccine, Gardasil^®^ and Gardasil 9^®^ [33], are neVLPs-based vaccines approved by the FDA. They are produced in *Saccharomyces cerevisiae*, an efficient expression system that is industrially scalable and cheaper than mammalian and insect cell systems. Despite these clear advantages, bacteria and yeast cells lack complex post-translational modifications (PTM) needed to produce some proteins, such as the highly glycosylated SARS-CoV-2 S protein [75,151]. Thus, the choice of the best expression system could be a determinant for protein/VLP production, even considering neVLPs. Cervarix^®^ is another HPV neVLP-based vaccine [152] which is highly immunogenic [153] and effective [154,155] against HPV types 16 and 18, which are the main serotypes that cause cervical cancer [148]. The Cervarix^®^ vaccine is produced using insect cells infected with recombinant baculovirus [40,156,157].

An up-and-coming SARS-CoV-2 neVLP-based vaccine candidate is the Novavax NVAX-CoV2373 (Figure 5E) that consists of S2P protein (1–1273) locked in the prefusion state and expressed using a baculovirus/insect cell system [32]. Later, the recombinant S protein is incorporated into polysorbate 80 detergent. This vaccine is formulated in combination with Matrix-M^®^ adjuvant [32]. The NVAX-CoV2373 vaccine was shown to be immunogenic and safe [35,36] and conferred 89.7% protection against symptomatic cases [158]. Using a different approach, the ABNCoV2 (Figure 5F) is a SARS-CoV-2 neVLPs-based vaccine candidate from the Radbound University and AdaptVac. A SpyTag-SpyCatcher was used to display the S RBD in a bacteriophage scaffold system [159]. The ABNCoV2 is another highly immunogenic candidate and induced high titers of nAbs in mice [159]. This vaccine candidate is currently in phase 1/2 clinical trials (NCT04839146).

Although these different approaches show great potential for developing a SARS-CoV-2 VLPs-based vaccine, some improvements could be made to design broad-spectrum vaccines against other coronaviruses of interest. A pancoronavirus vaccine is an exciting strategy against zoonotic coronaviruses that represent a threat to humans and may soon be responsible for the next pandemic [160,161].

It is already known that SARS-CoV-2 vaccines do not generate significant broadly neutralizing antibodies against other family members, supporting the need for a functional pancoronavirus vaccine [162]. Cohen and colleagues designed different mosaic nanoparticle vaccines based on the S RBD from several human and animal (bat and pangolin) sarbecoviruses, including the SARS-CoV-2 RBD (Figure 5G) [34]. Each vaccine induced strikingly higher cross-neutralizing antibodies than the SARS-CoV-2 S protein alone. Following the same idea, a Sortase A tagged ferritin-based VLPs conjugated to the SARS-CoV-2 S RBD (Figure 5H) was tested with alum adjuvants [163]. The protective immunity induced by this vaccine was robust against SARS-CoV-2 Alpha and Beta variants, as well as other human and bat sarbecoronaviruses. Interestingly, the cross-neutralizing antibody titers were more significant than the mRNA vaccine encoding the SARS-CoV-2 S protein. Despite this, the data suggest that mRNA and other S-based vaccines may afford some protection against other sarbecoronaviruses. Lastly, the candidate GBP510 (Figure 5I) is based on two rationally designed proteins, I53-50A and I53-50B [164,165], which self-assemble into trimers and pentamers, respectively. Combining these proteins forms a versatile 120 subunit scaffold, 28 nm wide, for the multivalent display of antigens linked to I53-50A subunits, such as SARS-CoV-2 RBD [156]. These particles were highly immunogenic in mice and resulted in 10-fold higher antibody titers at a 5-fold lower dose than the S2P protein alone [156]. In further preclinical trials with rhesus macaques, GBP510 was tested with different adjuvants, including alum, AS03, CpG, Essai O/W, and AS37. Although highly immunogenic against the original strain and the Alpha variant, a 4.5- to 16-fold reduction in neutralization was observed against the Beta variant [157]. GBP510 is already in phase I/II clinical trials (NCT04742738 and NCT04750343) sponsored by SK Bioscience Co. LTD and the Coalition for Epidemic Preparedness Innovations (CEPI).

## 5. Conclusions

As the cases of COVID-19 continue to grow and variants of SARS-CoV-2 emerge, the need for easily adaptable vaccine platforms remains one of the major goals in the pursuit of eradicating the disease. VLPs are an attractive platform for developing vaccines against infectious diseases such as COVID-19, due to their impressive versatility and immunological applications. The main advantage of using this vaccine platform is that VLPs can closely resemble the native virus in their size, shape and antigen display. Additionally, VLPs can be adapted to contain more than one antigen of interest, a very important feature to produce efficient and broad vaccines. 

Several VLP-based vaccines have been tested in human clinical trials since the approval of the Hecolin^®^, Cervarix^®^, Gardasil^®^ and Gardasil 9^®^ vaccines. These vaccines consolidated this platform, and opened doors for its use to produce vaccines against several diseases, including COVID-19. Here, we discussed nine VLP-based vaccines that have been developed against SARS-CoV-2, among which five are already undergoing clinical trials. The main antigenic component of these vaccines is the S protein, specifically the RBD, which is involved in viral entry and antibody neutralization, especially in its closed conformation. By displaying multiple full-length Spikes or only its RBD on the VLPs surfaces, especially on neVLPs, these vaccines have shown high immunogenicity in clinical trials, generating neutralizing antibodies, and inducing cell-mediated responses. However, we discussed the importance of considering additional structural proteins as antigens in eVLPs-based vaccines, as they lead to more complex and native-like structures, and also generate broad-spectrum immunity against variants and other coronaviruses. As seen with other available vaccines, rational and adaptable vaccine development directly lead to a reduction of hospitalizations, deaths and spread. The recent advances in VLP production and structural vaccinology provide this platform with all the needed features to produce novel, potent, and broad-spectrum coronavirus vaccines that could help in the fight against current and future pandemics.

## Figures and Tables

**Figure 1 vaccines-09-01409-f001:**
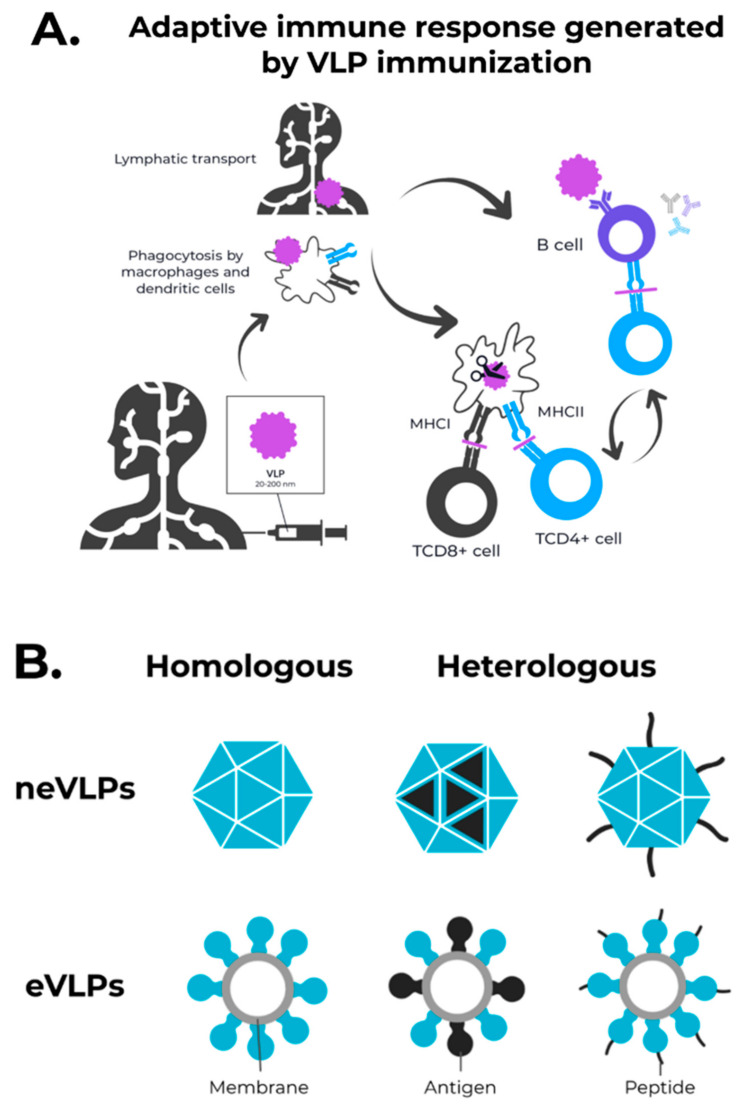
The adaptive immune response generated by VLPs immunization and VLPs classification. (**A**) After immunization, VLPs are phagocytized by dendritic cells or macrophages. Then, they are carried out to lymphatic vessels, where the antigenic regions will be processed and presented by class II MHC molecules (CD4+ T cells) and, through cross-presentation, by class I (CD8+ T cells). Immunological pathway activation by immunization with VLPs will activate robust cellular (cytokines) and humoral (B cell-antibodies) immune responses. (**B**) VLPs are classified as nonenveloped (neVLPs) or enveloped VLPs (eVLPs) based on the absence or presence of a lipidic membrane, respectively. These particles can also be classified as homologous or heterologous VLPs according to their composition. Homologous VLPs are assembled using proteins from the native pathogen only (blue), and heterologous VLPs can be assembled using proteins or peptides from different sources (black and blue).

**Figure 2 vaccines-09-01409-f002:**
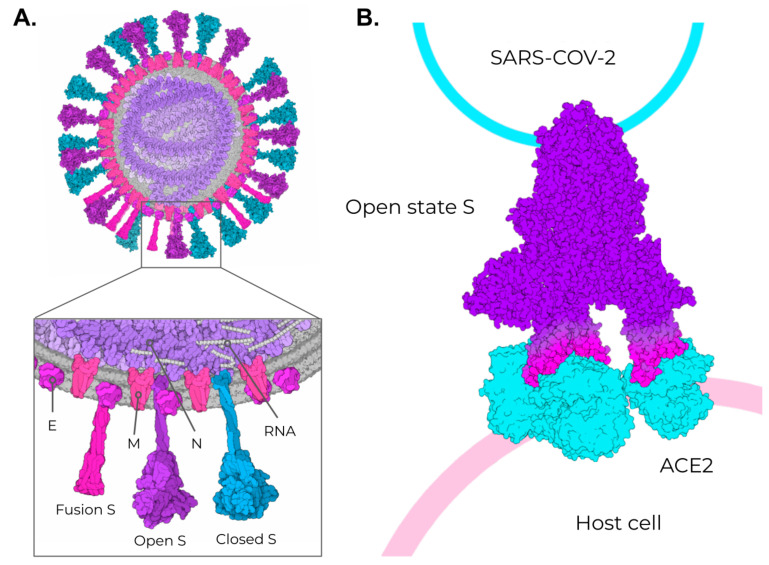
SARS-CoV-2 structural proteins and the different states of the Spike protein. (**A**) Schematic representation of the SARS-CoV-2 viral particle. The structure of the SARS-CoV-2 viral particle is composed of four structural proteins: Membrane (M), Envelope (E), Nucleocapsid (N), and Spike (S). The S protein is found in two different states on viral particles: open state (minor population) and closed state (major population). In addition, during the membrane fusion process (host cell entry), the S protein can be found in the fusion state (fusion S). (**B**) Schematic representation of the binding of open-state S (PDB ID 7498) to the ACE2 receptor present in the host cell. The illustrations were made in free software (CellPaint 2.0 [91] and 3D Protein Imager [92]). The binding figure was made using the crystal structure of ACE2 bound to Spike available at the Protein Data Bank (PDB ID 7A98).

**Figure 3 vaccines-09-01409-f003:**
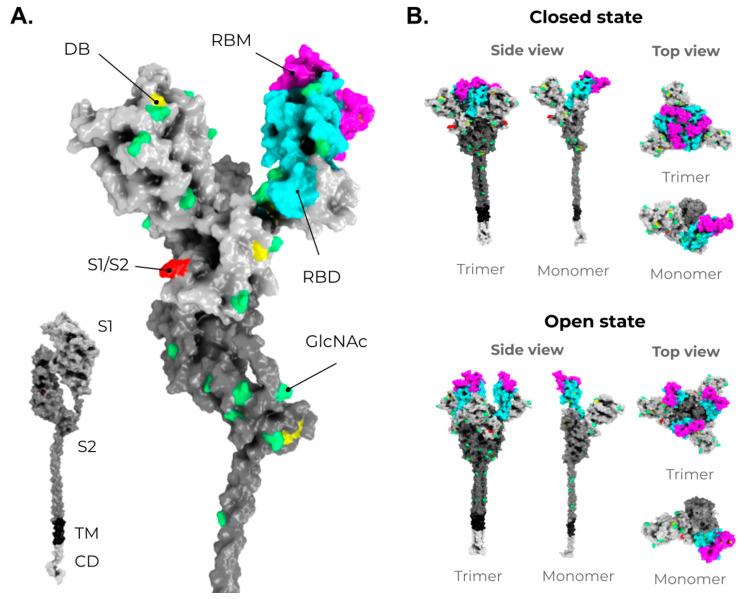
Structure and domain organization of the SARS-CoV-2 Spike (S) protein. (**A**) The S structure comprises a cytoplasmic domain (CD, white), a transmembrane domain (TM, black), and an ectodomain, which is divided into two subunits, S1 (gray) and S2 (dark gray). The magnification shows the several disulfide bridges (DB, yellow) and the glycosylation sites (GlcNAc, green) through the S protein ectodomain. It is highlighted in red, the S1/S2 interface. The receptor-binding domain (RBD, in cyan) and the receptor-binding motif (RBM, magenta) are also shown in S1. (**B**) As mentioned in Figure 2, the S protein shows two conformers on viable viruses (closed and open state). The upper panel shows the S protein in the closed state (trimeric and monomeric state). The bottom panel shows the S protein in the open state (trimeric and monomeric state). Illustrations were made in PyMol [110] using the wild-type structures available from Zhang et al. [107,111].

**Figure 4 vaccines-09-01409-f004:**
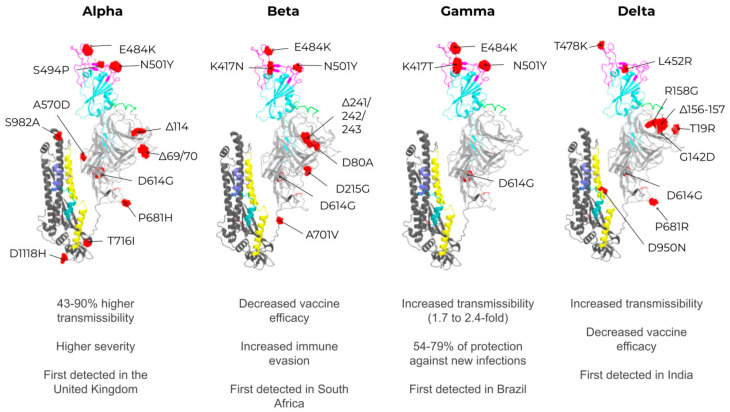
Mapping mutations of SARS-CoV-2 variants of concern (VOCs) and phenotypes. Red: mutations; Cyan: receptor-binding domain (RBD); Magenta: receptor binding motif (RBM); Light gray: S1; Dark gray: S2; Yellow: Heptad repeat 1; Green cyan: fusion peptide 1; Slate: Fusion peptide 2; Green: Signal peptide. Illustrations were made in PyMol [110] using resources from Zhang et al. [107,111].

**Figure 5 vaccines-09-01409-f005:**
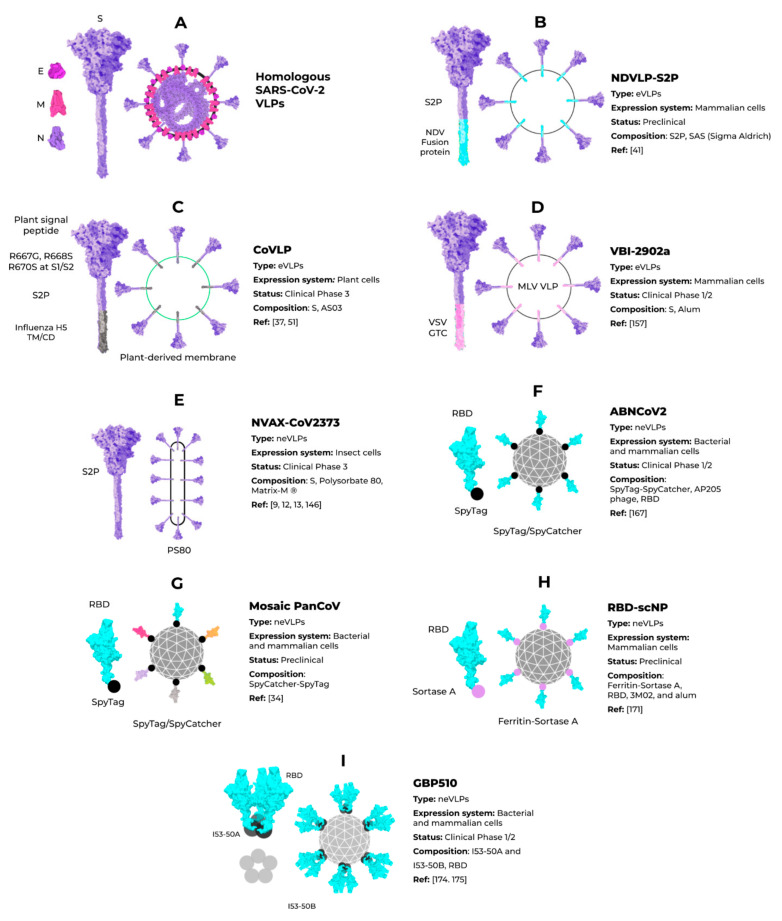
Enveloped and nonenveloped VLPs against SARS-CoV-2.

**Table 1 vaccines-09-01409-t001:** Summary of WHO COVID-19 approved vaccines.

Name	Platform	Adjuvant	Dosage	Efficacy *	References
Coronavac (Sinovac)	Inactivated	Alum	2 doses	83.5% (95% CI, 65.4–92.1)	[8,9,10,11]
BBIBP-CorV (Sinopharm)	Inactivated	Alum	2 doses	72.8% (95% CI, 58.1–82.4)	[12,13]
BBV152-Covaxin (Bharat Biotech)	Inactivated	Alum	2 doses	77.8% (95% CI, 65.2–86.4)	[14,15]
AZD1222–Vaxzevria (Oxford/AstraZeneca)	Viral vector	No	2 doses	74.0% (95% CI, 65.3–80.5)	[16,17,18]
Covishield (Oxford/AstraZeneca formulation)	Viral vector	No	2 doses	74.0% (95% CI, 65.3–80.5)	[16,17,18]
Ad26.COV2.S (Johnson &Johnson-Janssen)	Viral vector	No	1 dose	66.9% (95% CI, 59.0–73.4)	[19,20,21,22]
mRNA-1273 (Moderna)	mRNA	No	2 doses	94.1% (95% CI, 89.3–96.8)	[23,24]
BNT162b-Comirnaty (Pfizer/BioNTech)	mRNA	No	2 doses	95% (95% CI, 90.3–97.6)	[25,26]

* Against symptomatic COVID-19.

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
