# Peer review of "VLP-Based COVID-19 Vaccines: An Adaptable Technology against the Threat of New Variants"

_vaccines, 2021, doi:10.3390/vaccines9121409_

Round 1

Reviewer 1 Report

The recent COVID-19 virus pandemic has caused substantial global financial lose as well as high morbidity and mortality. Vaccination has been accepted as an efficient means for controlling he disease worldwide. So far, several different vaccines have been developed against the infection including those containing VLPs. The authors have explained different types of these VLP-based vaccines in this review. The authors could include a sub-heading eg “Mechanisms of VLPs immunogenicity” and explain the mechanisms of VLP immunogenicity and include Figure 1 in there. This section could compare the immunogenicity of the available COVID-19 vaccines

Minor comments

Line24. “SARS-CoV-2 antigens, the causative agent of COVID-19” does not sound right. Antigens are not the causative agents.

Line82-83. The sentence is misleading. The mature virion does not contain nonstructural proteins.

L141. VOC is explained in this line, but it appeared earlier in the text.

L171-172. “either in the condition of chronic infections or in people who have had Covid-19 infection”. Is this sentence aims to describe two different groups of people? 

Author Response

We would like to thank you for the valuable suggestions. All our responses (in bold) to the comments and suggestions made by the reviewers (normal text) may be found below. We hope the modified manuscript may now be acceptable for publication in Vaccines.

The recent COVID-19 virus pandemic has caused substantial global financial losses as well as high morbidity and mortality. Vaccination has been accepted as an efficient means for controlling the disease worldwide. So far, several different vaccines have been developed against the infection including those containing VLPs. The authors have explained different types of these VLP-based vaccines in this review. The authors could include a sub-heading eg “Mechanisms of VLPs immunogenicity” and explain the mechanisms of VLP immunogenicity and include Figure 1 in there. This section could compare the immunogenicity of the available COVID-19 vaccines

Thank you for your suggestions. We included a table with all the vaccines approved by WHO in this new version. We also included a paragraph discussing the immunogenicity of the VLP vaccines, which we referred to the figure 1A. We think that these and the following suggestions enriched our paper. 

Minor comments

Line24. “SARS-CoV-2 antigens, the causative agent of COVID-19” does not sound right. Antigens are not the causative agents.

We appreciate and agree with this. We deleted the word "antigens" in this new version. Thank you.

Line82-83. The sentence is misleading. The mature virion does not contain nonstructural proteins.

Indeed, we made a mistake here and corrected it. We excluded the "mature virion" and referred to the nonstructural proteins in the viral genome. Thank you.

L141. VOC is explained in this line, but it appeared earlier in the text.

We appreciate this correction and excluded the acronym explanation from this line in this new version. Thank you.

L171-172. “either in the condition of chronic infections or in people who have had Covid-19 infection”. Is this sentence aims to describe two different groups of people?  

Thank you for commenting on this. It was really confusing. We were describing the possible mechanism by which some variants could have originated during chronic covid-19 infections in immunocompromised patients.

Reviewer 2 Report

I commend the authors on a thorough, extremely well-written, and superbly organized review on this relevant topic. 

My only minor suggestion is to clearly indicate that the information on M, N, E, and S proteins detailed in lines 82-97 also comes from previous studies of other coronaviruses. As it currently reads, it gives the first impression that all the information presented was discovered just recently while studying SARS-CoV-2.

Author Response

We would like to thank you for the valuable suggestions. All our responses (in bold) to the comments and suggestions made by the reviewers (normal text) may be found below. We hope the modified manuscript may now be acceptable for publication in Vaccines.

I commend the authors on a thorough, extremely well-written, and superbly organized review on this relevant topic. 

My only minor suggestion is to clearly indicate that the information on M, N, E, and S proteins detailed in lines 82-97 also comes from previous studies of other coronaviruses. As it currently reads, it gives the first impression that all the information presented was discovered just recently while studying SARS-CoV-2. Clearly indicate that the information on M, N, E, and S proteins detailed in lines 82-97 also comes from previous studies of other coronaviruses. As it currently reads, it gives the first impression that all the information presented was discovered just recently while studying SARS-CoV-2. 

Thank you for pointing this out. We revised and correct the paragraph to pass the correct idea about other coronaviruses.

Reviewer 3 Report

The manuscript number 1432800 entitled “VLP-based COVID-19 vaccines: an adaptable technology against the threat of new variants” presents a review of the search for better, more efficacious vaccines simultaneously as the virus increasingly produces variants of concern (VOCs), focusing a special attention in the virus-like particles technology. This manuscript is interesting and addresses a very current topic, considering that COVID-19 pandemic affected humanity worldwide, bringing out incomputable human and financial losses and VOCs have demonstrating convergent evolution, associated with increased transmissibility and immune evasion. However some questions should be explored to improve the manuscript before to be considered for publication.

  1. In the topic 2 or in a subtopic should be systematized the vaccines already approved against COVID-19, and some in ongoing clinical studies to refer the nature of each vaccine (which molecule is used – mRNA, DNA, VLPs, viral…), if it is used delivery system, adjuvants… to point out the advantages and disadvantages in terms of action mode, safety, efficacy, stability, storing conditions, production time and costs. A Table can be useful to schematize the main information.
  2. Also in relation to the previous suggestion, some points can be explored in the discussion section, which is very simple and poor at the moment.
  3. To identify and highlight the efforts that are being made to adapt the approved vaccines or in clinical trials to cover the VOCs, in order to compare with what is proposed in this work with VLPs, which will be useful also to consolidate the discussion section.
  4. The topics 3 and 4 should be to the same target, or SARS-COV-2 or COVID-19, uniformize please.
  5. The vaccines described in these topics (3 and 4) can be summarized in a Table to be more perceptible the differences.
  6. In the same line of the description of Gardasil and Cervarix VLPs vaccines, the authors can also include the Gardasil 9 (the more recent approved HPV vaccine).  
  7. The topic Discussion according the current topics considered should be number 5.

Author Response

We would like to thank you for the valuable suggestions. All our responses (in bold) to the comments and suggestions made by the reviewers (normal text) may be found below. We hope the modified manuscript may now be acceptable for publication in Vaccines. Please, see the attachment with the tracking on Word.

The manuscript number 1432800 entitled “VLP-based COVID-19 vaccines: an adaptable technology against the threat of new variants” presents a review of the search for better, more efficacious vaccines simultaneously as the virus increasingly produces variants of concern (VOCs), focusing a special attention in the virus-like particles technology. This manuscript is interesting and addresses a very current topic, considering that COVID-19 pandemic affected humanity worldwide, bringing out incomputable human and financial losses and VOCs have demonstrating convergent evolution, associated with increased transmissibility and immune evasion. However some questions should be explored to improve the manuscript before to be considered for publication.

In the topic 2 or in a subtopic should be systematized the vaccines already approved against COVID-19, and some in ongoing clinical studies to refer the nature of each vaccine (which molecule is used – mRNA, DNA, VLPs, viral…), if it is used delivery system, adjuvants… to point out the advantages and disadvantages in terms of action mode, safety, efficacy, stability, storing conditions, production time and costs. A Table can be useful to schematize the main information.

Thank you for your suggestion. It was a great idea to compile some of the major characteristics of the available COVID-19 vaccines.  This table now provided  in the new version of this paper. 

Also in relation to the previous suggestion, some points can be explored in the discussion section, which is very simple and poor at the moment.

Indeed, in this version, we discussed some of these points in the discussion section. Thank you.

To identify and highlight the efforts that are being made to adapt the approved vaccines or in clinical trials to cover the VOCs, in order to compare with what is proposed in this work with VLPs, which will be useful also to consolidate the discussion section.

Thank you for your suggestion. Despite some efforts are being done to adapt the current vaccines to emerging VOCs, we decided to focus only on VLPs vaccines and the strategies that could be used to produce better vaccines using this platform.

Topics 3 and 4 should be to the same target, or SARS-COV-2 or COVID-19, uniformize, please.

Thank you for pointing this out. We have corrected and uniformized the names of the sections in this new version.

The vaccines described in these topics (3 and 4) can be summarized in a Table to be more perceptible the differences.

We appreciated this suggestion. However, as we already illustrated these vaccines in Figure 5, we chose to not summarize them in a table.

In the same line of the description of Gardasil and Cervarix VLPs vaccines, the authors can also include the Gardasil 9 (the more recent approved HPV vaccine).  

Thank you for suggesting this. We included the Gardasil 9 along with the other vaccines in this new version.

The topic Discussion according to the current topics considered should be number 5.

Thank you for pointing this out. We corrected and changed it to the number 5 in this new version.

Round 2

Reviewer 3 Report

In table 1, it is missing the first one approved vaccine, similar to Moderna it is missing “Pfizer-BioNTech” also based on mRNA.

Author Response

We would like to thank you for the valuable suggestion. Our response (in bold) to the comment (normal text) may be found below. We hope the modified manuscript may now be acceptable for publication in Vaccines.

"In table 1, it is missing the first one approved vaccine, similar to Moderna it is missing “Pfizer-BioNTech” also based on mRNA." We have inserted it and the complete table is provided below. Thank you.